# Discrete choice experiment to investigate preferences for psychological intervention in cardiac rehabilitation

Gemma E Shields ,[1] Adrian Wells,[2,3] Stuart Wright,[1] Caroline M Vass,[1,4] Patrick Joseph Doherty ,[5] Lora Capobianco,[3] Linda M Davies[1]

[1]Manchester Centre for Health Economics, The University of Manchester, Manchester, UK
[2]School of Psychological Sciences, The University of Manchester, Manchester, UK
[3]Greater Manchester Mental Health NHS Foundation Trust, Manchester, UK
[4]RTI Health Solutions, Manchester, UK
[5]Department of Health Sciences, University of York, York, UK

**Correspondence to**
Gemma E Shields;
gemma.shields@manchester.ac.uk

## ABSTRACT

**Objective** Cardiac rehabilitation (CR) is offered to people who recently experienced a cardiac event, and often comprises of exercise, education and psychological care. This stated preference study aimed to investigate preferences for attributes of a psychological therapy intervention in CR.

**Methods** A discrete choice experiment (DCE) was conducted and recruited a general population sample and a trial sample. DCE attributes included the modality (group or individual), healthcare professional providing care, information provided prior to therapy, location and the cost to the National Health Service (NHS). Participants were asked to choose between two hypothetical designs of therapy, with a separate opt-out included. A mixed logit model was used to analyse preferences. Cost to the NHS was used to estimate willingness to pay (WTP) for aspects of the intervention design.

**Results** Three hundred and four participants completed the DCE (general public sample (n=262, mean age 47, 48% female) and trial sample (n=42, mean age 66, 45% female)). A preference for receiving psychological therapy was demonstrated by both samples (general population WTP £1081; 95% CI £957 to £1206). The general population appeared to favour individual therapy (WTP £213; 95% CI £160 to £266), delivered by a CR professional (WTP £48; 9%% CI £4 to £93) and with a lower cost (β=−0.002; p<0.001). Participants preferred to avoid options where no information was received prior to starting therapy (WTP −£106; 95% CI −£153 to −£59). Results for the location attribute were variable and challenging to interpret.

**Conclusions** The study demonstrates a preference for psychological therapy as part of a programme of CR, as participants were more likely to opt-in to therapy. Results indicate that some aspects of the delivery which may be important to participants can be tailored to design a psychological therapy. Preference heterogeneity is an issue which may prevent a 'one-size-fits-all' approach to psychological therapy in CR.

## STRENGTHS AND LIMITATIONS OF THIS STUDY

⇒ The study is the largest known study on preferences for psychological therapy in cardiac rehabilitation programmes and it demonstrates that people would prefer to receive psychological therapy and there are aspects of the design and delivery that can be easily adapted to suit preferences.

⇒ The discrete choice experiment was informed by comprehensive rounds of patient and public involvement to outline included attributes and levels; however, the number of attributes is limited and subsequently some attributes of interest to professionals in this area may have been left out.

⇒ The study examines the preferences of two groups (general population and people with cardiac rehabilitation experience); however, selection bias and the sample size limit the conclusions of the study.

reduce morbidity and improve quality of life.[1] Furthermore, current evidence suggests that it is often cost-effective.[2] In the UK, around 90 000 people start CR annually.[3][4] Until the COVID-19 pandemic, participants most frequently accessed group-based supervised CR at a healthcare centre (2019 figures) though this has reduced (by 36%) during COVID-19.[3][4] The burden related to cardiovascular disease is increasing globally, with prevalent cases doubling between 1990 (271 million cases) and 2019 (523 million).[5] With this rise in cases, and a push for increased uptake as part of the NHS long-term plan, it is expected that the number of people accessing CR programmes will grow.[6]

CR programmes are typically comprised of sessions around exercises, education and psychological care.[1] Around 30% of people have symptoms of anxiety and 20% of people have symptoms of depression when they enter CR in the UK.[3] A qualitative study in the UK found that patients and nurses value psychological support being offered in CR though

## INTRODUCTION

Cardiac rehabilitation (CR), a supervised programme of care offered to people following a cardiac event, has been demonstrated to

the specific design and delivery affects feasibility due to resource constraints.[7] A review found that psychological interventions added to exercise-based CR were associated with reduced symptoms of depression and reduced cardiac morbidity.[8] Therefore, CR programmes are well placed to provide psychological interventions that offer advantages over standard care.

Discrete choice experiments (DCE) are a commonly used quantitative research method for preference elicitation in healthcare.[9 10] In a DCE, choices are made between hypothetical scenarios summarised using key attributes to represent an intervention or service.[9 10] They are based on the assumption that an in individual's valuation of an intervention/service will vary according to their preferences for levels of those described attributes.[11]

To date, preference research related to CR has focused on exercise and education activities.[12–17] The existing evidence base is methodologically heterogeneous, the CR interventions described using different attributes according to the research of the study.[12–17] Preference heterogeneity is a common theme across the identified papers, suggesting that a one-size-fits-all approach is unlikely to be suitable.[12–17] A systematic review of preference research for the treatment of anxiety and depressive disorders identified 11 relevant studies, noting significant heterogeneity in the design of studies.[18] The authors concluded that preferences for outcome, process and cost attributes differed and that clinicians and decision-makers should be aware of the attributes that may be important to patients. It was noted that aligning psychological treatment to patient preferences may help to increase adherence and subsequently the efficacy of treatment. A prior systematic review discussed the usefulness of DCEs in developing psychological interventions.[19]

The PATHWAY Programme funded under the UK NIHR Programme Grants for Applied Research (RP-PG-1211-20011), aims to improve access to more effective psychological interventions for patients attending CR who present with symptoms of depression and/or anxiety.[20–23] A multicentre, two-arm, single-blind, randomised controlled trial was conducted to compare the clinical-effectiveness and cost-effectiveness of group-based metacognitive therapy (MCT) plus CR to CR alone.[21] This found that group MCT+CR significantly improved depression and anxiety compared with CR alone.[21] There are many different potential designs and delivery choices for the implementation of psychological care, including MCT within the CR pathway.

A stated preference survey using a DCE was used to explore preferences for different characteristics of clinic-based psychological interventions added to the CR pathway. Furthermore, the existence of preference heterogeneity within and among the samples recruited was explored. This information can help services target improvements in CR to the aspects that are most important to current and future potential participants of cardiac rehabilitation.

## METHODS
The study used a DCE to examine preferences for psychological therapy in CR. The study received ethical approval from the NHS by Preston Research Ethics Committee: REC Reference 14/NW/0163.

### Attributes and levels
The choice of attributes and levels for inclusion in the DCE were decided following several rounds of comprehensive patient and public involvement (PPI) group discussion and feedback which is reported separately.[24] In brief, the PATHWAY PPI group were recruited from patient networks and had experience of heart disease, anxiety and/or depression, or being a carer of someone with one or more of these conditions. Five sessions were held with the PPI group to design the survey: starting with an introduction to the topic of preference research and covering refining the research question, selecting attributes and levels, designing and refining survey materials and discussing ideas for recruitment. Initial prompts for potential attributes were provided to initiate group discussion and were based on expert opinion, the potential design of psychological therapy (from the PATHWAY trial), qualitative interview feedback and a review of the literature. The separate paper reports in more details the benefits and challenges of PPI to inform DCE design.[24]

The DCE design (attributes and levels) were constructed following interactive work with the PPI group and wider trial team. Each hypothetical scenario included five attributes: (1) psychological intervention type (focusing on group or individual formats), (2) professional providing therapy, (3) information providing prior to therapy, (4) location and (5) cost to the NHS. The full characteristics and levels included in the discrete choice experiment design are included in table 1.

The first attribute focused on the delivery of the psychosocial therapy, which was determined to be key from the perspective of the research team and the PPI group. The first level (peer support) is included to reflect the minimum level of conversation-based intervention and reflects the PPI group's positive feedback about peer support. As discussed, the DCE is part of the PATHWAY programme of work which considers the use of MCT within CR. As part of the trial, MCT is being delivered in a way that reflects the second level (group psychological therapy where you are not required to share personal concerns/experiences) though it could also be delivered on an individual basis (the fourth level). The third level reflects alternative therapy structures that may require participants to disclose personal information. The second attribute concerns the healthcare practitioner who delivers the therapy, as therapies can be delivered by a range of staff (which has resource use and cost implications) and people may feel more or less comfortable with different practitioners. The PPI group noted that their decision to attend sessions could be affected by whether they knew what to expect from the therapy. As this would be easy to address in practise, the third attribute was

**Table 1** Characteristics and levels

| Attribute | Levels |
|---|---|
| 1. Psychological intervention to be received alongside your standard cardiac rehabilitation programme | 1. Peer group support that provides non-specific support and advice<br>2. Group psychological therapy where you are not required to share detailed information about personal concerns/experiences<br>3. Group psychological therapy where you may be required to share detailed information about personal concerns/experiences<br>4. Individual psychological therapy |
| 2. The person who provides the psychological therapy | 1. Occupational therapist trained to deliver psychological therapy<br>2. Cardiac rehabilitation professional trained in delivery of psychological therapy<br>3. Healthcare professional trained in delivery of psychological intervention, no background in cardiac rehabilitation or psychology<br>4. Clinical psychologist |
| 3. The information given to you prior to accepting and starting treatment that gives you an idea of what to expect from the therapy | 1. No information provided<br>2. A printed leaflet of information<br>3. An overview of the therapy from a healthcare provider with a chance to ask questions<br>4. An overview of the therapy from a healthcare provider with a chance to ask questions and a printed leaflet |
| 4. Location you need to visit to attend psychological therapy sessions | 1. Primary care (GP surgery)<br>2. Community care (NHS clinic in the community)<br>3. Outpatient (clinic at a hospital)<br>4. Tertiary care (specialist/University hospital) |
| 5. Additional cost to the NHS | 1. £0 per person per course of therapy<br>2. £500 per person per course of therapy<br>3. £1000 per person per course of therapy<br>4. £2000 per person per course of therapy |

GP, general practitioner; NHS, National Health Service.

developed to explore this. The final qualitative attribute focuses on the location of therapy, which was included as some members of the group felt this would be important, with examples provided in bracket to clarify care settings. Note the location here refers to the delivery of therapy and at this stage patients will have transitioned through other care settings (eg, for the diagnosis of cardiac symptoms and psychological symptoms). Finally, one quantitative attribute was included which considered the additional cost to the NHS. This enabled the study to estimate willingness to pay (WTP) for aspects of delivery of psychological therapy.

### Experimental design
A fractional factorial design was chosen, to reduce the number of scenarios by selecting a sample of possible combinations that covers the combinations and effects of interest, using a published design catalogue (http://neilsloane.com/oadir/oa.16.5.4.2.txt) and modulo arithmetic.[25 26] Participants were asked to choose their preferred scenario from two hypothetical options, and then whether they would choose this scenario or no psychological therapy (opt-out). An opt-out was included as this is more reflective of real life, in which not all potential participants choose to attend CR.[3 25]

An example question is provided in figure 1.

Questionnaires included four sections; basic sociodemographic details (age, gender, ethnicity, employment status and education level), the EQ-5D-5L for health status measurement, the DCE choice sets and a final section for the participants to collect any feedback. EQ-5D values (utility values) were estimated using the crosswalk mapping algorithm, in line with current guidance from the National Institute for Health and Care Excellence.[27 28] The online survey was developed using Lighthouse Studio Software V.9.8. The survey was prepilot tested with the PPI group and refined by further discussion, prior to seeking ethics approval.

### Recruitment
The study aimed to recruit a range of participants, including a trial group and a general population sample. The largest recruited group were a sample of the general public aged 18 and over from the UK, recruited via Dynata (formally Research Now), an online commercial survey sample provider. Eligible panel members (aged over 18 and living in the UK) received a link to the online survey and were reimbursed through the panel incentives. The remaining two groups were recruited from the Group-MCT trial and included participants and healthcare professionals. Trial participants were sent a paper copy of the survey materials, as well as an optional online survey link, and were invited once trial follow-up had completed. Full trial inclusion and exclusion criteria are reported separately but in brief, participants had to be adults aged over 18, with a competent level of English language comprehension, referred to CR services and presenting with symptoms of anxiety and/or

**Figure 1** Example question.

depression.[21 29] Healthcare professionals involved in the Group-MCT trial were invited to participate and sent an online survey link. Trial participants and trial healthcare professionals were offered a high street gift voucher to reimburse their time.

## Sample size

Estimating sample sizes needed for a DCE is difficult as a wide range of factors need to be considered, including the DCE design (eg, the number of attributes, levels and choice sets) and expected preference heterogeneity.[30] There are also many practical constraints (time and budget). An estimate can be derived based on the number of choice sets in the fractional factorial design.[31] This estimated a minimum sample of 96 responses that was needed for each choice set. However, it was recognised that this 'calculation' is a rule of thumb and that a higher number of responses will be required if there is significant heterogeneity in the sample of participants.

## Patient and public involvement

There was extensive PPI to formulate the survey design, materials and to interpret the results. The PPI group worked alongside the research team (including a PPI coordinator) over five sessions. Note, PPI is reported as a case study in a separate paper considering the value of PPI in stated preference research.[24]

## Analysis

The DCE was analysed using individual choice responses as the dependent variable in the model.[32] Random utility theory assumes that a participant chooses the option that provides the highest overall utility or value to them, which in a DCE involves interpreting the information described in each choice set. Therefore, the coefficients for each characteristic will indicate the direction of preference for that characteristic. A conditional logit, using maximum likelihood estimation, was used in the first instance.[33 34] This was initially conducted using an internal pilot of 100 participants recruited from the general public sample for a preliminary analysis, which confirmed that the DCE design could be analysed and indicated respondents generally understood the questions.

Once all responses were collected, mixed logit models were analysed for each group to allow for preference heterogeneity within the group. A Swait and Louviere plot was then created to identify potential scale heterogeneity and differences in preferences.[35 36]

The marginal rates of substitution (MRS) for each characteristic were calculated to estimate willingness to trade off among characteristics. The MRS for each characteristic was estimated by dividing the coefficient for that characteristic by the inverse of the coefficient for the NHS cost characteristic. CIs for the MRS values were calculated using the delta method. This allowed the estimation of the relative cost participants were willing to accept for the different aspects of design (WTP).

## RESULTS

In total, 304 participants completed the stated preference survey. Of these, 262 were members of the general public (recruited via Dynata). Trial participants (n=265) were invited, with six responding online and 32 responding by post (a response rate of 14%). Finally, 14 healthcare practitioners involved in the PATHWAY study were invited, with four completing the survey (a response rate of 29%).

An overview of the participant characteristics is provided in table 2.

Typically, compared with the general population sample recruited via an online panel, the trial sample were more likely to be older, White, in unpaid employment and with were less likely to have General Certificate of Secondary Education or equivalent qualifications. As would be expected, due to their age and trial status, they were more experienced with regard to cardiac problems, psychological therapy and were more likely to have close ones affected by cardiac problems.

The mean EQ-5D value, which represents health status, was 0.80 (SD 0.22; range −0.59 to 1) for the general population sample and 0.69 (0.24; range 0.14 to 1) for the trial sample. This is lower than the population norms for people of a similar age; which is 0.862 for the group aged 45–54 and 0.784 for the group aged 65–74.[37]

**Table 2** Participant characteristics

| Characteristic | General population sample (n=262) | Trial sample (n=42)* | Total (n=304) |
|---|---|---|---|
| Gender | | | |
| Female | 127 (48.47%) | 19 (45.24%) | 146 (48.03%) |
| Male | 132 (50.38%) | 23 (54.76%) | 155 (50.99%) |
| Prefer not to answer or missing | 3 (1.15%) | 0 (0.00%) | 3 (0.99%) |
| Age | | | |
| Mean age (SD) | 47.43 (16.29) | 66 (10.02) | 50 (16.83) |
| Prefer not to answer or missing | 1 (0.38%) | 0 (0.00%) | 1 (0.33%) |
| Ethnicity | | | |
| White | 225 (85.88%) | 41 (97.62%) | 266 (87.50%) |
| Mixed/multiple ethic groups | 4 (1.53%) | 0 (0.00%) | 4 (1.32%) |
| Asian/Asian British | 18 (6.87%) | 0 (0.00%) | 18 (5.92%) |
| Black/Africa/Caribbean/Black British | 10 (3.82%) | 0 (0.00%) | 10 (3.29%) |
| Other ethnic group | 1 (0.38%) | 1 (2.38%) | 2 (0.66%) |
| Prefer not to answer or missing | 4 (1.53%) | 0 (0.00%) | 4 (1.32%) |
| Employment | | | |
| Paid employment | 163 (62.21%) | 10 (23.81%) | 173 (56.91%) |
| Unpaid employment | 57 (21.76%) | 30 (71.43%) | 87 (28.62%) |
| Unemployed | 36 (13.74%) | 2 (4.75%) | 38 (12.50%) |
| Prefer not to answer or missing | 6 (2.29%) | 0 (0.00%) | 6 (1.97%) |
| Education level | | | |
| GCSE or equivalent | 246 (93.89%) | 31 (73.81%) | 277 (91.12%) |
| No GCSE or equivalent | 13 (4.96%) | 10 (23.81%) | 23 (7.57%) |
| Prefer not to answer or missing | 3 (1.15%) | 1 (2.38%) | 4 (1.32%) |
| Healthcare professional working in cardiac rehabilitation | | | |
| Yes | 8 (3.05%) | 4 (9.52%) | 12 (3.95%) |
| No | 249 (95.04%) | 38 (90.48%) | 287 (94.41%) |
| Missing | 5 (1.91%) | 0 (0.00%) | 5 (1.64%) |
| Prior experience of cardiac problems | | | |
| Yes with experience of cardiac rehabilitation | 22 (8.40%) | 35 (83.33%) | 57 (18.75%) |
| Yes with prior offer of cardiac rehabilitation not undertaken | 5 (1.91%) | 0 (0.00%) | 5 (1.64%) |
| Yes with no prior offer or experience of cardiac rehabilitation | 15 (5.73%) | 1 (2.38%) | 16 (5.26%) |
| No | 217 (82.82% | 5 (11.90%) | 222 (73.03%) |
| Prefer not to answer or missing | 3 (1.15%) | 1 (2.38%) | 4 (1.32%) |
| Family or someone close to the participant affected by cardiac problems | | | |
| Yes | 120 (45.80%) | 27 (64.29%) | 147 (48.36%) |
| No | 122 (46.56%) | 14 (33.33%) | 136 (44.74%) |
| Do not know | 13 (4.96%) | 0 (0.00%) | 13 (4.28%) |
| Prefer not to answer or missing | 7 (2.67%) | 1 (2.38%) | 8 (2.63%) |
| Prior experience of psychological therapy | | | |
| Yes | 60 (22.90%) | 14 (33.33%) | 74 (24.34%) |
| No | 194 (74.05%) | 27 (64.29%) | 221 (72.70%) |
| Prefer not to answer or missing | 8 (3.05%) | 1 (2.38%) | 9 (2.96%) |

*Trial participants and healthcare practitioners involved in the study were invited. Due to the sample size limitations, these groups were pooled, justified according to their level of experience with CR and group-based intervention formats.
GCSE, General Certificate of Secondary Education.

## Preferences for psychological therapy delivery

The preliminary results using conditional logit are provided in the supplementary material (online supplemental table 1). To assess the presence of heterogeneity, a Swait and Louviere plot was created, which confirmed the presence of potential scale and preference heterogeneity (included in the online supplemental figure 1). Subsequently mixed logit models were used for analysis. The comparison of the models was conducted by using MRS values which control for scale differences between the groups.

The trial sample failed to reach the target sample size for recruitment, which may in part be due to the impact of the COVID-19 pandemic. Furthermore, seven trial participants failed to respond to all choice set questions and were subsequently excluded from the analysis. Therefore, results for this sample must be interpreted with caution. The results of the mixed logit models are presented in table 3.

Both samples have a positive coefficient for the constant indicating that participants would be more likely to opt-in to therapy in CR (vs opting out), in the general population sample, the WTP for therapy provision of £1081. The general population sample were slightly more likely to opt-in to therapy compared with the trial sample (93% vs 86%).

The results indicate that the general population prefers individual therapy, as shown by the (statistically significant) positive coefficient for this level. Coefficients are negative for the group-based options, and size increases with the perceived level of psychological therapy and involvement. This suggests that the general public perhaps do not feel comfortable with sharing their feelings as part of group therapy. Regarding the professional delivering therapy, there is a statistically significant positive coefficient for a cardiac rehabilitation professional and direction is favourable for clinical psychologist though not significant. Compared with the remaining healthcare professionals, it could be assumed that the sample favoured healthcare practitioners who appeared more specifically qualified to the disease/condition area (ie, cardiac rehabilitation or anxiety and/or depression) rather than healthcare practitioners with broader activity areas. Participants favoured having some degree of information provided prior to therapy, with the strongest preference (statistically significant) for an overview from a healthcare provider plus a chance to ask questions. The location attribute is challenging to interpret though statistically significant results were shown for primary care (negative) and community care (positive) which perhaps indicates a preference for community-based care outside of a general practice setting. Participants favoured options with lower costs incurred to the NHS.

Given the sample size reached for the trial participants, there are very few attributes/levels with statistically significant results. Similar to the general population sample, a preference for staff from a cardiac rehabilitation background was demonstrated and participants did not want

to receive no information prior to therapy and favoured having an overview from a healthcare professional with a chance to ask questions. Furthermore, trial participants also preferred a lower cost to the NHS and showed a preference for therapy provision vs no therapy provision). No significant results were identified for therapy design (ie, peer support, group-based or individual), or location.

The SD reveal the degree of heterogeneity for each attribute level included in the survey. A significant SD implies that preferences varied systematically among the survey respondents. As shown in table 3, the SD for the levels: receiving group psychological therapy with sharing personal concerns/experiences, group psychological therapy without sharing, and NHS cost are statistically significant at the 5% level in the general population sample suggesting preference heterogeneity for these features. In the trial sample, significant SDs were found for the levels: receiving the intervention in group psychological therapy with sharing personal concerns/experiences, group psychological therapy without sharing and peer group support; delivery by a trained healthcare professional and clinical psychologist; information as an overview of the therapy from a healthcare provider with a chance to ask questions and a printed leaflet and NHS cost, suggesting significant preference heterogeneity among these respondents.

## DISCUSSION

A key finding of this study is that both samples demonstrated preferences for receiving psychological therapy as part of CR (ie, they were more likely to opt-in than to opt-out, irrespective of the therapy design). The results of this study suggested that participants may favour individual therapy, in contrast with group-based formats. Furthermore, receiving information prior to starting therapy was important, highlighting the importance of healthcare practitioners communicating effectively with patients. It is possible that by providing information prior to initiating therapy, participant preferences may change (eg, if they understand more about the potential strengths and limitations of therapy). Participants across both samples preferred therapy to be delivered by cardiac rehabilitation professionals and options with a lower cost to the NHS. Results for the location attribute were variable and challenging to interpret.

The DCE was informed by comprehensive rounds of PPI which are reported separately.[24] There were many benefits to this PPI engagement, most notably the practical insights offered by PPI group members, related to attribute selection, questionnaire design and recruitment. However, there were some challenges, such as the time needed to conduct PPI activities. Interestingly, the attribute that was included but had least agreement from the group (ie, some felt it was very important and others disagreed) was location.

As noted in the introduction, previous DCEs had heterogeneous designs and tended to focus on exercise

**Table 3** Mixed logit results

| Attribute/level | General population sample (n=262) | | | Trial sample (n=42) | | |
|---|---|---|---|---|---|---|
| | Coef. (SE; p) | WTP GBP £ (95% CIs) | Significant SD† | Coef. (SE; p) | WTP GBP £ (95% CIs) | Significant SD† |
| Psychological intervention to be received | | | | | | |
| Peer group support | −0.012 (0.054; 0.817) | −5 (−50 to 40) | No | 0.082 (0.144; 0.570) | 92 (−224 to 407) | Yes |
| Group psychological therapy (without sharing personal concerns/experiences) | −0.208 (0.056; <0.001*) | −89 (−136 to −42) | Yes | 0.048 (0.147; 0.745) | 54 (−270 to 378) | Yes |
| Group psychological therapy (sharing personal concerns/experiences) | −0.279 (0.063; <0.001*) | −119 (−173 to −65) | Yes | −0.124 (0.147; 0.399) | −139 (−461 to 183) | Yes |
| Individual psychological therapy | 0.500 (0.064; <0.001*)* | 213 (160 to 266) | No | −0.006 (0.185; 0.975) | −2 (−157 to 152) | No |
| The person who provides the psychological therapy | | | | | | |
| Occupational therapist | −0.023 (0.054; 0.672) | −10 (−55 to 35) | No | 0.026 (0.124; 0.834) | 29 (−245 to 303) | No |
| Cardiac rehabilitation professional | 0.113 (0.053; 0.032*) | 48 (4 to 93) | No | 0.589 (0.131; <0.001*) | 662 (338 to 987) | No |
| Healthcare professional trained in delivery of psychological intervention | −0.145 (0.055; 0.008*) | −62 (−108 to −16) | No | −0.700 (0.166; <0.001*) | −787 (−1180 to −393) | Yes |
| Clinical psychologist | 0.055 (0.054; 0.314) | 23 (−22 to 68) | No | 0.085 (0.162; 0.601) | 36 (−99 to 172) | Yes |
| The information given to you prior to accepting and starting therapy | | | | | | |
| No information provided | −0.249 (0.055; <0.001*) | −106 (−153 to −59) | No | −0.362 (0.140; 0.010*) | −407 (−718 to −96) | No |
| Printed leaflet of information | −0.028 (0.054; 0.607) | −12 (−57 to 34) | No | 0.072 (0.137; 0.601) | 81 (−219 to 380) | No |
| Overview of the therapy from a healthcare provider with a chance to ask questions | 0.190 (0.053; <0.001*) | 81 (36 to 126) | No | 0.318 (0.122; 0.009*) | 357 (82 to 632) | No |
| Overview of the therapy from a healthcare provider with a chance to ask questions and a printed leaflet | 0.087 (0.053; 0.102) | 37 (−7 to 82) | No | −0.028 (0.143; 0.847) | −12 (−131 to 108) | Yes |
| Location you need to visit to attend psychological therapy sessions | | | | | | |
| Primary care | −0.135 (0.055; 0.015*) | −57 (−103 to −11) | No | 0.215 (0.146; 0.142) | 241 (−75 to 558) | No |
| Community care | 0.150 (0.052; 0.004*) | 64 (20 to 108) | No | 0.022 (0.122; 0.856) | 25 (−245 to 295) | No |
| Outpatient | −0.074 (0.055; 0.178) | −32 (−78 to 14) | No | −0.026 (0.127; 0.836) | −30 (−309 to 250) | No |
| Tertiary care | 0.058 (0.054; 0.279) | 25 (−20 to 70) | No | −0.211 (0.148; 0.154) | −90 (−213 to 34) | No |
| Cost | | | | | | |

Continued

**Table 3** Continued

| Attribute/level | General population sample (n=262) | | | Trial sample (n=42) | | |
|---|---|---|---|---|---|---|
| | Coef. (SE; p) | WTP GBP £ (95% CIs) | Significant SD† | Coef. (SE; p) | WTP GBP £ (95% CIs) | Significant SD† |
| NHS cost | −0.002 (0.000; 0.000)* | – | Yes | −0.001 (0.000; 0.000)* | – | Yes |
| Therapy provision | | | | | | |
| Alternative specific constant‡ | 2.540 (0.149; 0.000)* | 1081 (957 to 1206) | Yes | 1.783 (0.419; 0.000)* | 759 (409 to 1109) | Yes |

*Statistical significance (p<0.05).
†SD is based on the normal distribution assumed for each attribute level in the model. These were interpreted using a statistical significance p<0.05.
‡This constant represents people's preferences for some form of therapy in CR (specifically one with the mean effect for each of the qualitative attributes and no cost) vs receiving no therapy.
CR, cardiac rehabilitation; GBP, British Pounds; NHS, National Health Service; WTP, willingness to pay.

and educational sessions. Though our study has some novel findings in relation to the previous literature, like the other studies, we also identify that preference heterogeneity is an additional complication.[12–17] Kjær et al found that younger women were more likely to value individual CR meetings (not therapy).[14] Our study found that preferences appeared stronger for individual therapy, but perhaps larger sample sizes would help to identify groups with differing preferences. A comparison between samples is limited due to the sample size of trial participants. Analysis indicates that there is likely to be scale and preference heterogeneity across the surveyed groups. Preference heterogeneity, within samples, has been noted to be an issue in previous DCEs in this area.[4–8] Prior research in CR has identified multiple patient characteristics associated with CR uptake, including age, gender, ethnicity, relationship status, comorbidities and social deprivation.[38–40] Further research should aim to recruit larger and more varied samples, for example targeting poorly served and under-represented groups, to more thoroughly assess how preferences differ by group. DCE targeted in these populations has potential to inform adaptations in service delivery to increase access and health across the wider population accessing CR.

The COVID-19 pandemic is likely to have affected the DCE, both in terms of the preferences elicited and the sample size. Two of the attributes included in the design are highly likely to be affected by changes in preferences and behaviours during the pandemic. First, both samples appeared to prefer individual therapy delivered one-to-one by a healthcare professional. While this may reflect a preference for privacy, it could also be that participants were wary of group situations during national lockdown due to risks associated with infection. Second, location preferences may have been altered if participants had been keen to avoid busy areas or perceived certain settings to be associated with a higher risk of contracting COVID-19 (eg, hospital settings). The trial participant response rate is also likely to have been impacted by the pandemic, as many of the target population with have been isolating due to risk factors (ie, age and health conditions) and may have been unwilling to leave the house to postsurveys back. However, this may have been affected by other factors too. In particular, it had been some time since trial follow-up had completed and participants may have moved or passed away.

This study adds to the evidence base by providing information on the acceptability of psychological therapy in CR and on preferences for the characteristics of delivery. There are limitations of the research, most notably selection bias and the sample size and lack of significant results (especially in the trial sample). As noted above, further research could address this. Participants were able to leave free-text comments at the end of the survey and some noted that it felt repetitive (especially trial participants), future studies could consider blocking designs to reduce participant burden. There were two formats used which differed between the surveyed groups; the general

public group completed an online survey, whereas the trial participants predominantly completed paper surveys. As survey mode can influence error variance, this may have impacted differences in the elicited preferences between groups. DCEs can only include a limited number of attributes, the wider literature provides some interesting additional ideas for exploration. For example, Boyde *et al* considered whether CR sessions are delivered in or outside of working hours which would be interesting for therapy options too.[12] A separate DCE by this research team focused on whether CR participants preferred home or clinic-based formats of psychological therapy in CR.[41] This pilot study identified that home-based formats appeared to be preferred however, this was not statistically significant. Furthermore, some of the included characteristics could be defined in different ways, for example, location could be defined by ease of transport and other environmental factors. Further research is needed to explore preferences across all modes of delivery. Related to psychological treatments, fear of stigma may affect whether patients access available psychological therapy or pharmacological treatment (ie, opt in).[42] Further preference research could investigate the impact of, and the link between fear of stigma and opt-in, as well as differences in preferences for psychological therapy or pharmacological treatment options.

## CONCLUSION

The study highlights a need to consider psychological therapy as part of a comprehensive package of CR, as participants were most likely to opt-in to therapy than they were to opt-out, suggesting they felt there is a need for additional intervention. Results also indicate some aspects of the delivery may be important to participants (such as the format, level of information and who provides therapy) which can be tailored to design a psychological therapy as part of CR that reflects preferences. Preference heterogeneity is an issue which may prevent there from being a 'one-size-fits-all' approach to psychological therapy delivery in CR. Further research should investigate this in more detail with larger sample sizes and efforts to target subgroups of the population with greater need who are currently poorer served.

**Acknowledgements** We are very grateful to the PATHWAY PPI members (past and present), and PATHWAY PPI co-ordinators Tracey Williamson, Carolyn Gamble and Lindsey Brown for assisting with PPI activities that informed this work, as well as the wider PATHWAY team.

**Contributors** AW and LMD conceived the idea for the research project. AW is the chief investigator in the PATHWAY project. AW, GES, CMV, LMD, PJD and LC designed and facilitated the experiment in part or in full. SW and GES conducted the analysis. GES drafted the first version of the manuscript and all authors contributed to the subsequent versions. All authors read and approved the final manuscript. GES acted as guarantor for the paper.

**Funding** This paper presents independent research funded by the National Institute for Health Research (NIHR) under its Programme Grants for Applied Research (PGfAR) Programme (Grant Reference Number RP-PG-1211-20011).

**Disclaimer** The views expressed are those of the authors and not necessarily those of the NIHR or the Department of Health.

**Competing interests** None declared.

**Patient and public involvement** Patients and/or the public were involved in the design, or conduct, or reporting or dissemination plans of this research. Refer to the Methods section for further details.

**Patient consent for publication** Not required.

**Ethics approval** This study involves human participants. This study was reviewed and given a favourable ethical opinion for conduct in the NHS by Preston Research Ethics Committee: REC Reference 14/NW/0163. Participants gave informed consent to participate in the study before taking part.

**Provenance and peer review** Not commissioned; externally peer reviewed.

**Data availability statement** Anonymised data are available upon reasonable request from the corresponding author.

**ORCID iDs**
Gemma E Shields http://orcid.org/0000-0003-4869-7524
Patrick Joseph Doherty http://orcid.org/0000-0002-1887-0237

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
