## [Reviewer comments · BMJ Open]

ARTICLE DETAILS

TITLE (PROVISIONAL)	A discrete choice experiment to investigate preferences for psychological intervention in cardiac rehabilitation
AUTHORS	Shields, Gemm; Wells, Adrian; Wright, Stuart; Vass, Caroline M.; Doherty, Patrick Joseph; Capobianco, Lora; Davies, Linda

VERSION 1 – REVIEW

REVIEWER	Fornaro, M Columbia University,, New York State Psychiatric Institue
REVIEW RETURNED	04-Jun-2022

GENERAL COMMENTS	A discrete choice experiment is a quantitative method increasingly used in healthcare to elicit preferences from participants (patients, payers, commissioners) without directly asking them to state their preferred options. This is a nice and innovative approach in the field. Did the authors also consider fear or stigma against psychopharmacological agents as a positive predictive factor in the preference of psychological therapy vs. other methods? Even people without a conclusive diagnosis of depression or anxiety may nonetheless experience psychological discomfort or sleep disturbances after a major cardiac event and are seldom prescribed benzodiazepines or antidepressants. I think this issue could be briefly mentioned in the discussion. Besides, the present piece reads very well, and I agree with the authors about the acknowledged limitation of the potential selection bias.
---

REVIEWER	Komasi, Saeid Kermanshah University of Medical Sciences
REVIEW RETURNED	07-Jun-2022

GENERAL COMMENTS	Dear Editor of BMJ Open Thanks for referring the article entitle "A discrete choice experiment to investigate preferences for psychological intervention in cardiac rehabilitation" to me. In general, the article covers an important area and provides valuable information through an appropriate methodological structure. I think this article has the potential to be published after a minor revision. My comments are visible below: In the abstract, the authors can refer to the number of people in the two groups of general population and trial sample. Mentioning
---

	the mean age and sex of the sample can also be helpful. It is better to mention the abbreviations such as NHS in full first. Keywords should be selected from the Mesh. They can also be sorted alphabetically. Introduction Well organized. However, some of these sentences need reference. Add information about context (locations and environments) how to measure psychological symptoms to the methods section can be useful. Best Regards
--	---

VERSION 1 – AUTHOR RESPONSE

Comment	Revision
Reviewer 1	
A discrete choice experiment is a quantitative method increasingly used in healthcare to elicit preferences from participants (patients, payers, commissioners) without directly asking them to state their preferred options. This is a nice and innovative approach in the field.	Thank you for your comment (no revision required).
Did the authors also consider fear or stigma against psychopharmacological agents as a positive predictive factor in the preference of psychological therapy vs. other methods? Even people without a conclusive diagnosis of depression or anxiety may nonetheless experience psychological discomfort or sleep disturbances after a major cardiac event and are seldom prescribed benzodiazepines or antidepressants. I think this issue could be briefly mentioned in the discussion. Besides, the present piece reads very well, and I agree with the authors about the acknowledged limitation of the potential selection bias.	Thank you to the reviewer for raising this point. Fear of stigma is an interesting topic, which we think will likely affect opt in (i.e. if people favour psychological therapy over pharmacological agents) and opt out (i.e. if the fear of stigma is great enough to prevent patients from seeking any psychological care). In relation to this comment it would be interesting for future preference research to compare preferences for pharmacological options versus psychological options in the cardiac population. A discussion point has been added to address this “Related to psychological treatments, fear of stigma may affect whether patients access available psychological therapy or pharmacological treatment (i.e., opt in) [42]. Further preference research could investigate the impact of, and the link between fear of stigma and opt-in, as well as differences in preferences for psychological therapy or pharmacological treatment options.”
Reviewer 2	
In general, the article covers an important area and provides valuable information through an appropriate methodological	Thank you for your comment, we have made minor amends in line with the other comments received.

structure. I think this article has the potential to be published after a minor revision.	
---	--

VERSION 2 – REVIEW

REVIEWER	Fornaro, M Columbia University,, New York State Psychiatric Institue
REVIEW RETURNED	27-Aug-2022

GENERAL COMMENTS	Thank you for incorporating my suggested editing.
---

REVIEWER	Komasi, Saeid Kermanshah University of Medical Sciences
REVIEW RETURNED	04-Sep-2022

GENERAL COMMENTS	Dear Editor of BMJ Open I reviewed the manuscript again and thanks to the authors for applying my comments in the text. The manuscript can be accepted after a minor revision: As I mentioned earlier, the keywords should be listed in alphabetical order. Regards
--